# The Role of Sex Differences in Bone Health and Healing

**DOI:** 10.3390/biology12070993

**Published:** 2023-07-12

**Authors:** Elena Ortona, Maria Teresa Pagano, Lavinia Capossela, Walter Malorni

**Affiliations:** 1Center for Gender Specific Medicine, Istituto Superiore di Sanità, 00161 Rome, Italy; mariateresa.pagano@iss.it; 2Institute of Pediatrics, Fondazione Policlinico A. Gemelli IRCCS—Università Cattolica Sacro Cuore, 00168 Rome, Italy; laviniacapossela@gmail.com; 3Center for Global Health, Università Cattolica del Sacro Cuore, 00168 Rome, Italy; walter.malorni@unicatt.it

**Keywords:** bone health, sex differences, gender differences, fracture healing

## Abstract

**Simple Summary:**

Fracture healing is a complex process that includes a framework of events triggered by tissue injury. Clinical experience with bone healing revealed a series of cellular and biochemical actors encompassing the repair mechanisms in human beings. However, the different responses of individuals in this scenario are still a matter of debate. We analyze herein in some detail the disparity between men and women in this process. Based on the literature, we suggest that different mechanisms could underlie bone healing in men and women and that the role of estrogen could be pivotal in delayed fracture repair observed in women.

**Abstract:**

Fracture healing is a long-term and complex process influenced by a huge variety of factors. Among these, there is a sex/gender disparity. Based on significant differences observed in the outcome of bone healing in males and females, in the present review, we report the main findings, hypotheses and pitfalls that could lead to these differences. In particular, the role of sex hormones and inflammation has been reported to have a role in the observed less efficient bone healing in females in comparison with that observed in males. In addition, estrogen-induced cellular processes such as autophagic cell cycle impairment and molecular signals suppressing cell cycle progression seem also to play a role in female fracture healing delay. In conclusion, it seems conceivable that a complex framework of events could contribute to the female bias in bone healing, and we suggest that a reappraisal of the compelling factors could contribute to the mitigation of sex/gender disparity and improve bone healing outcomes.

## 1. Introduction

Sex- and gender-specific medicine deals with the detection and study of the disparity between males and females or men and women in biology and medicine. For years this issue was completely neglected by investigators, but, in the last years, epidemiological evidence first and clinical data later clearly demonstrated that the incidence, the prevalence, the course and sometimes the symptoms themselves of many diseases clearly displayed that sex (biological) or gender (sociocultural such as lifestyles including nutritional habits) should be considered in all aspects of diseases (Table 1). Cardiovascular diseases, immune and autoimmune diseases, and oncological or infectious diseases showed impressive sex/gender disparity if the dataset was analyzed after stratification of the results considering this issue. Hence, recent studies demonstrated that from diagnosis to therapeutic intervention, the relevance of gender-specific medicine could provide useful insights into the development of tailored prevention strategies and the appropriateness of the cures. On the basis of these works and the recommendations to pay more attention to sex and gender issues from institutional agencies (the US National Institutes of Health proposed considering sex as a biological variable in 2016), medical specialties other than those reported above began to investigate if gender medicine could be of interest. Among these also, orthopedics has recently been involved in this matter with the aim of clarifying if bone fracture and healing could be a further field of investigation [1,2,3,4,5,6]. 

In fact, orthopedics and bone research and clinics have provided some interesting clues in recognizing the determinants of sex disparity in various diseases. For example, it has been observed that sex and gender-related differences may influence the outcome of patients undergoing total hip arthroplasty. Female patients seem to require specific care rules either in the preoperative or intraoperative and postoperative phases [7]. Proximal humerus fractures are more common in the elderly female population, together with other fragility fractures such as proximal femur and hip fractures. However, there is evidence that the mortality rate in patients with proximal humerus fractures is higher in the male population [8,9,10].

A further relevant issue came from the studies on cartilage repair. It was observed that males and females differ in cartilage degeneration and repair. Stem cell therapy could contribute to these differences. In particular, the sex of the stem cell donor as well as that of the recipient, seems to play a role [11]. However, the fields of interest appear to be extremely diverse. For instance, the outcomes following anterior cruciate ligament reconstruction clearly display significant sex disparity [12] and, even in pediatric age, differences between females and males in anatomy, hormone and neuromuscular patterns lead to a higher vulnerability of females to knee injury; in particular, for patellofemoral pain syndrome and anterior cruciate ligament rupture [13]. These are just a few examples of completely different applications of gender medicine in clinical practice that underscore how the complex interplay among biological, physiologic, and social issues should be more deeply investigated in the different fields of orthopedics. 

Notwithstanding, the ideal management for complex bone fractures also represents a significant unresolved matter in orthopedics and related specialties from a mechanistic point of view. Fracture healing is a multistage process that includes several complex steps starting after tissue injury. In particular, bone healing can be characterized by three partially overlapping phases: the inflammatory phase, the repair phase, and the remodeling phase. Even though understanding of the biological processes and molecular signals that coordinate fracture repair has advanced, the causes of variability observed in fracture repair are poorly understood. From a general point of view, body weight might play a crucial role in bone regeneration processes, influencing the healing process since local tissue tensions are important for callus tissue development. Elevated strains at the fracture area induce mesenchymal cells to form fibrous tissue, whereas low-stress conditions lead to the generation of osseous tissue. At intermediate stresses, mesenchymal cells differentiate into chondrocytes and induce the development of cartilaginous callus [14,15,16].

Epidemiological studies report a very high number of patients with hip fractures worldwide (more than 1.6 million) with significant differences between men and women, e.g., concerning spinal fractures (29.3/1000 for women and 13.6/1000 for men) [17,18]. In fact, the main sites of fracture are the hip and the spine, with the former being very common due to fragility that can be present in elderly people with osteoporosis, which can be one of the main causes of disability. In particular, it was noted that 22% of women and 33% of men die in the first year after hip fracture [19], so this represents a critical issue either for patients or, in view of hospitalization needs, also for the public health services. Furthermore, the quality of life of these patients can be strongly impaired since they could suffer from spinal deformities [20], reduction of pulmonary function [21] changes and impairment of their daily activity, and, more generally, pain [22,23]. Fracture healing is a complex and long-term process, and osteogenesis and healing time can be influenced by several different factors (such as blood supply and/or inflammatory state). Failures in fracture healing are also detected in 5–10% of patients [22,24]. 

## 2. Sex Differences in Bone Health and Healing

Skeletal tissue displays sex differences in morphology and physiological function, which can have an impact on bone healing [25]. For example, men tend to have stronger and larger bones compared to women, which can make them more resilient to injury and less prone to fractures. Moreover, in females, the risk of developing osteoporosis sharply increases after menopause, while the occurrence of osteoporosis in men progressively rises with age [26], and this represents a fundamental issue. Hence, from a clinical point of view, these sex differences lead to an epidemiological gap not only in the occurrence and fracture risk but also in the patient management and clinical outcome [27]. 

Regarding bone fracture healing, some clinical studies reported that males show more rapid fracture healing. In contrast, women may have an increased risk for atrophic non-unions rather than hypertrophic non-unions, as observed in males [28,29]. By contrast, in other clinical studies, no influence of sex on fracture healing in specific fracture types has been observed [14,30,31].

In the elderly, men show higher post-operative complications and mortality after hip fractures, whereas women have a higher risk for developing non-unions after femoral neck fractures. To note, up to a third of patients with hip fractures can be totally disabled because of non-union [32]. In a prospective study of more than one thousand patients with intracapsular fractures of the femoral neck, a significantly higher incidence of non-union has been found in females in comparison to males [33]. 

## 3. Sex Hormones and Bone Healing

As a general rule, osteoblasts and osteoclasts are special cells that help bones to grow and develop. Osteoblasts form new bones and add growth to existing bone tissue. Conversely, osteoclasts dissolve old and damaged bone tissue that can be thus replaced with healthier cells created by osteoblasts. Hormones impact this key interplay influencing bone healing. Differences in sex hormone levels, their timing and activity, and the composition of the inflammatory milieu underlie variations in bone healing by sex. In particular, estrogen suppresses bone resorption by inducing osteoclast apoptosis and, on the other hand, promotes bone formation by increasing osteoblast survival [34,35]. For instance, estrogen inhibits osteoclast activity by regulating vascular endothelial growth factor (VEGF) production [36], an essential signal of importance for angiogenesis required for bone development.

Moreover, this hormone can also play a role in fracture regeneration, modulating the self-renewal of skeletal stem cells [36,37,38,39,40,41,42]. 

In a recent study to investigate how estrogens modulate bone regeneration, Andrew et al. [43] compared bone fracture healing between adult male and female mice. The authors observed the healing response to be significantly stronger in male than female mice. This corresponded to a higher frequency of skeletal stem cells (SSC) in the femora of male mice in comparison with the femurs of female mice of the same age and weight. 

In female mice, estrogen signaling modulated SSCs to mediate regeneration, whereas male SSCs did not react to estrogen. Estrogen acts directly on the SSC by up-regulating multiple skeletogenic pathways and appears to be necessary for self-renewal and differentiation of SSCs. These results also suggest a clinically applicable strategy to accelerate bone healing using localized estrogen hormone therapy. Estrogen also induces secretion of osteoprotegerin (OPG), which binds to an osteoclast differentiation factor called RANK-L (Receptor activator of nuclear factor kappa-Β ligand also known as tumor necrosis factor ligand superfamily member 11), leading to inhibition of osteoclast maturation [44]. However, it should be considered that the role of estrogen in bone growth and maturation is central in both males and females since, for example, estrogen deficiency due to estrogen receptor mutation or aromatase deficiency in males resulted in osteopenia and no epiphyseal closure [45,46].

The role of androgens has also been investigated. Testosterone has positive effects on bone metabolism in adult males decreasing insulin-like growth factor binding protein IGFBP-4 (belonging to a group of proteins transporting the insulin-like growth factor 1), which has inhibitory effects on osteoblast differentiation but also increasing IGFBP-2 and IGFBP-3, which instead stimulate this process [47]. Testosterone can also increase the expression of OPG, inducing the inhibition of osteoclast maturation [48]. However, the role of testosterone in OPG expression appears to be still controversial. In fact, 5α-dihydrotestosterone (DHT) seems to reduce OPG in a dose-dependent manner [49].

Venken et al. [50] suggested that androgens could play a role in the sexual dimorphism of bone growth and development. In particular, these authors showed that testosterone rescues orchiectomy-induced bone loss, confirming the importance of androgen receptor signaling in male skeleton regulation. Figure 1 outlines the main mechanisms played by estrogen and androgen.

Estrogen inhibits osteoclasts’ activity by regulating vascular endothelial growth factor (VEGF) production [36] and induces secretion of osteoprotegerin (OPG), leading to inhibition of osteoclast maturation [44]. Estrogen is able to prevent osteoblast apoptosis by inhibiting the decrease of Bcl-2 in osteoblasts. Testosterone decreases insulin-like growth factor binding protein IGFBP-4, which has inhibitory effects on osteoblasts and increases IGFBP-2 and IGFBP-3, which instead stimulate osteoblasts. Testosterone is also able to increase the expression of OPG, inducing the inhibition of osteoclast maturation [48]. 
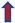

**= induction **
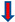

**= inhibition.**

## 4. Inflammatory Milieu and Sex Differences

After a fracture, immune cells such as polymorphonuclear leukocytes (PMNs), natural killer (NK) cells, mast cells and platelets are activated and initiated to produce cytokines/chemokines that recruit monocytes/macrophages to further play important roles throughout this process. The pro-inflammatory cytokines, including interleukin (IL)-1, IL-6, and tumor necrosis factor (TNF)-α, represent crucial factors not only during the early stages of bone fracture but also in the late repair and remodeling phases. A balanced immune response is crucial for successful bone healing since bone regeneration is modulated by local and systemic inflammatory responses [51,52,53]. It is well-known that B- and T-lymphocytes, as well as macrophages and neutrophils, could modulate fracture healing [54,55,56]. 

Haffner-Luntzer and co-workers [57] showed that estrogen deficiency influences the early inflammatory phase after a fracture. This may contribute to delayed fracture healing after estrogen depletion supporting the clinical relevance of delayed bone healing in postmenopausal patients with osteoporosis. The initial recruitment of inflammatory cells to the fracture callus was unaffected by estrogen depletion. However, in the absence of estrogen, prolonged recruitment and increased survival of neutrophils at the fracture site were observed on day 3 after fracture, although the mechanisms of interaction between estrogen and neutrophils remain unclear. Moreover, through TGF β, estrogen inhibits T cell proliferation and differentiation and INF-γ production. These events lead to the reduction of TNF production and osteoclastic activity [58]. At variance, estrogen deficiency increases the production of IL-12 and IL-18, enhancing T cell activation and TNF production and finally causing bone loss [59].

Some further insights derive from investigations carried out in animals. In particular, the mouse model has certain advantages. With respect to other species, e.g., rabbits, mice are economical and have a strong ability to resist infection and tolerate surgery, and the repair cycle is shorter and easier to manage [60]. Osipov et al. [10], using 3-month-old male and female C57BL/6J mice, observed sex differences in systemic bone loss after transverse femur fracture. One-day post-fracture, IL-6 and IL-1β were elevated in fractured mice of both sexes, but TNF-α was elevated in male fractured mice only [10].

During bone healing, phagocytic cells, such as macrophages, migrate at the site of damage to eliminate dead cells and debris. Macrophages also secrete cytokines and chemokines, which stimulate inflammation and promote angiogenesis [51,61,62]. These events happen immediately after fractures and, in a short time, generally a few days, decline and the regenerative processes start [63]. Three types of macrophages have been described: undifferentiated M0, pro-inflammatory M1, and anti-inflammatory M2. Recently, Nathan et al. [64] suggested that the process of bone healing is described by a preliminary pro-inflammatory phase mediated by M1, followed by an anti-inflammatory reaction determined by M2 [51,65]. M1 is also able to inhibit human mesenchymal stem cell (hMSC) development, whereas, on the contrary, M2 stimulates hMSC growth. To note, the crosstalk between macrophages and hMSCs varies to some extent between men and women [64] hence contributing to sex differences in fracture healing.

## 5. Cell Mechanisms 

Fine subcellular mechanisms related to sex disparity in bone healing are poorly understood. As a general rule, it has been suggested that differences in XX and XY cells could be observed in terms of differentiation, e.g., in neuronal stem cells [66] or muscle cells [67], as well as, more generally, in tissue repair [68] and development [69]. Concerning bone mass and osteoblast activity, it has been suggested that, at least in murine models, estrogen receptor (ER)-α in osteoblast progenitors and hypertrophic chondrocytes differentially contributes to bone mass regulation in male and female mice [70] or that male and female human osteoblast respond differently to orthopedic biomaterials [71]. In particular, they respond similarly to microstructures but exhibit sexual dimorphism in substrate-dependent responses to estrogen [71]. However, some work has been published more specifically devoted to the study of osteoblast differentiation from a sex-specific perspective. An excellent work studying craniosynostosis underlines the fact that sagittal and metopic synostosis have a male preponderance (3:1), whereas premature fusion of the coronal suture has a female preponderance (2:1). Then the authors investigate either the activity of alkaline phosphatase (ALP), indicating the formation of new bone, or bromodeoxyuridine (BrdU) incorporation indicating early stages of osteoblast differentiation and proliferation. They found striking sex-specific gene expression patterns and that transcripts related to osteoblast differentiation were differentially up- and down-regulated and correlated with ALP activity compared to controls [72].

Regarding cell death/cell survival regulatory mechanisms, significant differences were found between XX and XY cells in various histotypes. In particular, it was suggested that apoptotic cell death and autophagic cytoprotection could be induced differently in cells from males and females. The ability of XX cells to survive better than XY cells to exogenous injuring stressors has been detected in various cell types suggesting the idea of a higher apoptotic proneness of XY cells compared with a higher autophagic proneness of XX cells [73]. There are few investigations dealing with this issue on cells committed to bone healing. Recent work in mice and in vitro has shown that Bcl-2, an important regulator of apoptosis, could have a role in bone homeostasis and development, affecting bone phenotype regulated by estrogen. In fact, estrogen is able to prevent osteoblast apoptosis by inhibiting the decrease of Bcl-2 in osteoblasts. 

Studies with animals provide further important information in this field. Endoplasmic reticulum (ER) stress, which is related to apoptosis in several cell types, is associated with the expression of a cell cycle blocker called CHOP (CCAAT/enhancer binding protein homologous protein). Overexpression of CHOP in the bone microenvironment seems to impair the osteoblastic function leading to osteopenia with a sex disparity. CHOP deficiency alters BMD, bone microstructure and osteoblastogenesis, indicating that ER stress-related CHOP signaling suppresses cell cycle progression and may play an important role in bone formation in mice, especially in female mice. Further studies are needed to clarify whether the estrogenic signaling pathway could be involved in the observed sex differences in CHOP-mediated susceptibility to osteopenia [74].

A further point to be underlined concerns the cytoprotection mechanism of autophagy mentioned above. This is an important metabolic mechanism by which cells can get an energy supply for their survival. Hence, the disparity between XX and XY cells previously hypothesized [73,75] could be pivotal in explaining sex differences in several biological processes, including bone healing. In fact, it seems able to alleviate oxidative stress in two key bone cell types: osteoblasts and osteocytes. Once more, studies carried out in mice indicated that autophagic modulation in bone cells differs according to age, sex and cell type. In particular, the lowering of autophagy in female osteoblasts was associated with a higher oxidative imbalance playing a role in osteoporosis pathophysiology and suggesting that autophagy could be a new therapeutic target for osteoporosis in women [76].

## 6. Other Molecular Factors

Further actors have been suggested to be involved in sex differences in bone healing. Among these are β-catenin signaling, 5-Lipoxygenase (5-LO) and insulin-like growth factor 1 (IGF-1), a hormone similar in molecular structure to insulin which plays an important role in childhood growth and has anabolic effects in adults. Regarding the first, a study analyzing sex differences in fracture healing in C57BL/6J mice reported that male mice display more rapid fracture healing with more prominent cartilaginous callus formation [77]. The authors observed that male mice displayed significantly greater activation of osteoanabolic Wnt/β-catenin signaling, a family of proteins that play critical roles in embryonic development and adult tissue homeostasis, which might also contribute to more rapid bone regeneration. 

As concerns 5-LO, it has been observed that after an acute fracture, the inhibition or reduction of local 5-LO leads to augmented bone formation [78]. Furthermore, 5-LO catalyzes the development of leukotrienes, inflammatory mediators secreted by activated mast cells, from arachidonic acid [79]. Interestingly, 5-LO inhibitors have been observed to be more effective in females since androgens seem able to inhibit molecular mechanisms downstream of 5-LO in males [80].

IGF-1 also plays a crucial role in bone growth and healing. IGF-1 is produced mainly by the stimulation of growth hormone (GH) [81]. Interestingly, high levels of estrogen reduce serum IGF-1 concentration, whereas testosterone indirectly (by aromatization to estrogen) and directly induces the increase of IGF-1 [82,83,84], suggesting sexually dimorphic effects of IGF-1 on bone health and structure [85].

Additionally, other studies support the hypothesis that the delayed bone formation following fractures observed in female rats compared with male rats is in part due to a lower number of MSCs in female rats that must be considered [86,87]. 

Finally, differences between the bone marrow composition of men and women in relation to the development of osteoporosis have also been suggested [88,89]. In fact, the quality of the bone has been observed to be intimately related to the composition of the bone marrow (yellow bone marrow is made mostly of fat and contains stem cells that can become cartilage, fat, or bone cells). Several studies highlighted a correlation between the amount of fat in the bone marrow and bone fragility [90,91,92]. 

## 7. Pediatric Age and Sex Differences in Bone Healing

Regarding pediatric prepuberal age, no sex differences have been reported in bone mineral content (BMC) and BMD. On the other hand, during puberty, females had significantly higher BMC and BMD of the spine and the pelvis, whereas, at postpubertal age, males showed higher BMC and BMD than females [93].

In their recent study, Baxter-Jones et al. [94] found that girls matured approximately two years earlier than boys (11.8 vs. 13.4 years) but, on average, were shorter, had less lean mass and had greater fat mass. Moreover, there was a disconnection between the growth and the mineralization of bones in both sexes. Boys had greater bone mass and bone geometry. Diet and physical activity were crucial factors in obtaining optimal bone mass during adolescence in both sexes [94]. 

Gabel et al. [95] supported the idea that the accelerated periosteal apposition during adolescence was more evident in boys than in girls. On the other hand, girls experienced diminished endocortical resorption compared with boys. Furthermore, the same group [96] conducted a mixed longitudinal HR-pQCT study on sex differences and growth-related adaptations in bone from childhood to early adulthood, evaluating that there were no sex differences in CT-BMD. Greater bone size and strength in boys compared with those in girls may be advantageous, but boys’ consistently more porous cortices may subsidize their higher fracture incidence during adolescence. 

Bone in children may fail under compression; less initial stability and less callus formation are required to reach fracture healing [97]. 

Genes and hormones needed for the initial development of the skeleton are similar to those required for fracture healing. Hence, fracture healing processes are already ongoing in children, whereas in adults, these factors must be re-established, explaining the slower healing time in adults. Differently to that observed in adults, in children, no significant sex difference was observed regarding fracture healing.

## 8. Conclusions

Broadly, males tend to have more robust healing compared to females, at least in adult ages. Differences in sex hormones levels, composition of the inflammatory milieu, cellular and molecular mechanisms, but also lifestyles and behaviors underlie variations in bone healing by sex. Further in vitro and in vivo studies are needed to better understand the roles of sex and gender determinants on bone healing.

## Figures and Tables

**Figure 1 biology-12-00993-f001:**
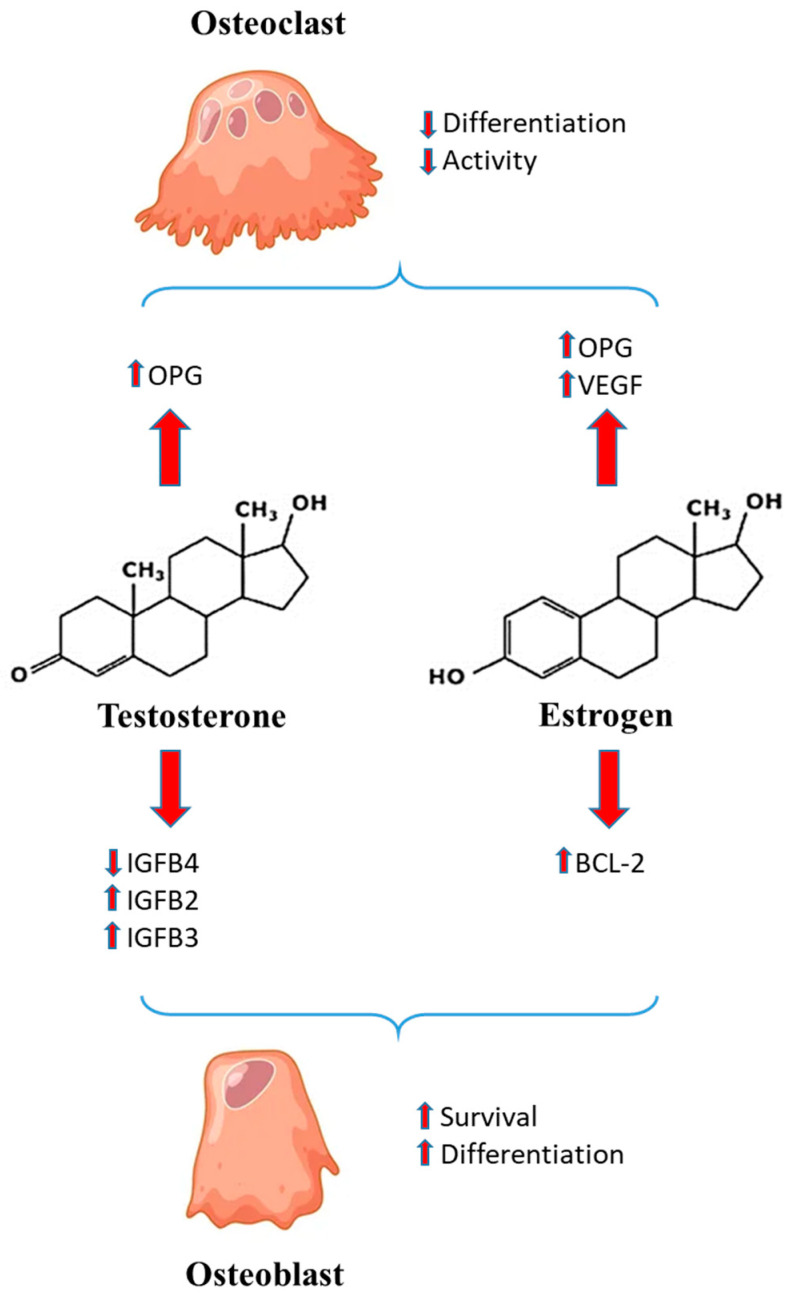
A schema showing the effects of estrogen and testosterone on differentiation, activity and survival of osteoblasts and osteoclasts.

**Table 1 biology-12-00993-t001:** Some examples of human pathologic conditions showing sex/gender differences in terms of incidence, course and clinical manifestations. A paradigmatic example for each pathology is reported.

Pathology	Sex/Gender Differences				
	Incidence	Course	Symptoms	Example	References
Cardiovascular diseases	Yes	Yes	Yes	Infarction	[1]
Neurodegenerative Diseases	Yes	No	No	Alzheimer	[2]
Autoimmune diseases	Yes	Yes	Yes	Lupus	[3]
Infectious diseases	Yes	Yes	No	Hepatitis B	[4]
Cancers	Yes	Yes	No	Melanoma	[5]
Respiratory diseases	Yes	No	No	Chronic obstructive pulmonary disease	[6]
Orthopedics	Yes	Yes	No	Hip arthroplasty, Hip, Femur, Humerus fractures	[7,8,9,10]

## Data Availability

Not applicable.

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
