# Peer review of "The Role of Sex Differences in Bone Health and Healing"

_biology, 2023, doi:10.3390/biology12070993_

Round 1
Reviewer 1 Report
This work summarized an extensive review on the effects of sex on bone health and healing from the aspects of sex hormone levels, inflammatory environment composition, cellular and molecular mechanism differences, etc. There are some limitations in this manuscript.
1, The abstract should be written concisely with some clearly conclusive sentences for the summary.
2, In Table 1, the author should summarize orthopedics pathology with varieties, such as fractures in head of femur, wrist, etc.
3, In the first paragraph of page 4, it was stated that the SSC activity of male mice was higher, but the reason was not explained. Moreover, the regulation of androgens hormones on bone was not well-described in this paper. And the difference of Bcl-2, CHOP and Apelin-13 on bone regulation in difference sex was not explained clearly.
4, The mechanisms under different sex hormones should be characterized with better logic, and with one or two diagrams for a better understanding.
5, The part of “Life styles, bone health and gender differences” is too simple. I suggest the author either delete this part, or write it more accurately with more literature supported.
Author Response
Comments and Suggestions for Authors
This work summarized an extensive review on the effects of sex on bone health and healing from the aspects of sex hormone levels, inflammatory environment composition, cellular and molecular mechanism differences, etc. There are some limitations in this manuscript.
1, The abstract should be written concisely with some clearly conclusive sentences for the summary.
AUTHORS: both the abstracts have been modified. They now include some conclusive sentences
2, In Table 1, the author should summarize orthopedics pathology with varieties, such as fractures in head of femur, wrist, etc.
AUTHORS: some variety of orthopedic pathology has now been provided in Table 1
3, In the first paragraph of page 4, it was stated that the SSC activity of male mice was higher, but the reason was not explained. Moreover, the regulation of androgens hormones on bone was not well-described in this paper. And the difference of Bcl-2, CHOP and Apelin-13 on bone regulation in difference sex was not explained clearly.
AUTHORS: this part of the work has deeply been revised in order to better explain these points. The sentences dealing with the possible implication of Apelin-13 have been omitted, too preliminary and complex to be explained.
4, The mechanisms under different sex hormones should be characterized with better logic, and with one or two diagrams for a better understanding.
AUTHORS: A simple diagram is now provided and the text revised accordingly.
5, The part of “Life styles, bone health and gender differences” is too simple. I suggest the author either delete this part, or write it more accurately with more literature supported.
AUTHORS: according to this reviewer concern this part of the work has been removed
Reviewer 2 Report
The authors highlight the sex difference in relation to the state of the bones, and in particular to their repair. The authors list a number of compelling reasons why bone repair in men and women occurs according to different physiological and molecular mechanisms. However, in my opinion, this difference is a direct consequence of the general difference between bones in women and men. The authors forgot to mention an important aspect that determines bone quality and that has been the subject of intense research in recent years, in relation to the development of osteoporosis. The quality of the bone is intimately related to the composition of the bone marrow (mainly composed of different types of fatty acids and water). Several studies have highlighted the correlation between the amount of fat in the bone marrow and bone fragility (D. Mattioli et al. BONE 2022; 164, 116514 https://doi.org/10.1016/j.bone.2022.116514; G. Di Pietro et al. Acad. Radiol. 2016; 23(3), Pages 273-283. https://doi.org/10.1016/j.acra.2015.11.009; Pino, Ana María, et al. Bone 2019; 118: 53-61. https://doi.org/10.1016/j.bone.2017.12.014). Some of these have also shown differences between the bone marrow of men and women in relation to the development of osteoporosis (W. Shen et al. Osteoporos Int. 2012; 23(9): 2293–2301. doi:10.1007/s00198-011-1873-x and G.P. Liney et al. JMRI 2007; 26(3) Pages 787-793. https://doi.org/10.1002/jmri.21072)
I, therefore, suggest authors add a paragraph to the manuscript that discusses the difference in bone recovery in relation to the different compositions of bone marrow in men and women. The relative amount of bone marrow fat changes with variations in physical activity and nutritional changes.
Author Response
The authors highlight the sex difference in relation to the state of the bones, and in particular to their repair. The authors list a number of compelling reasons why bone repair in men and women occurs according to different physiological and molecular mechanisms. However, in my opinion, this difference is a direct consequence of the general difference between bones in women and men. The authors forgot to mention an important aspect that determines bone quality and that has been the subject of intense research in recent years, in relation to the development of osteoporosis. The quality of the bone is intimately related to the composition of the bone marrow (mainly composed of different types of fatty acids and water). Several studies have highlighted the correlation between the amount of fat in the bone marrow and bone fragility (D. Mattioli et al. BONE 2022; 164, 116514 https://doi.org/10.1016/j.bone.2022.116514; G. Di Pietro et al. Acad. Radiol. 2016; 23(3), Pages 273-283. https://doi.org/10.1016/j.acra.2015.11.009; Pino, Ana María, et al. Bone 2019; 118: 53-61. https://doi.org/10.1016/j.bone.2017.12.014). Some of these have also shown differences between the bone marrow of men and women in relation to the development of osteoporosis (W. Shen et al. Osteoporos Int. 2012; 23(9): 2293–2301. doi:10.1007/s00198-011-1873-x and G.P. Liney et al. JMRI 2007; 26(3) Pages 787-793. https://doi.org/10.1002/jmri.21072)
I, therefore, suggest authors add a paragraph to the manuscript that discusses the difference in bone recovery in relation to the different compositions of bone marrow in men and women. The relative amount of bone marrow fat changes with variations in physical activity and nutritional changes.
AUTHORS: we are sorry for our inaccuracy. These interesting works and related items have been included in the revised version of the work
Round 2
Reviewer 1 Report
The author made some changes which improved the manuscript greatly. I recommend acceptance.
Reviewer 2 Report
The manuscript reports an excellent and useful work